# H_2_O_2_ Induces Association of RCA with the Thylakoid Membrane to Enhance Resistance of *Oryza meyeriana* to *Xanthomonas oryzae* pv. *oryzae*

**DOI:** 10.3390/plants8090351

**Published:** 2019-09-16

**Authors:** Qiong Mei, Yong Yang, Shenhai Ye, Weifang Liang, Xuming Wang, Jie Zhou, Chulang Yu, Chengqi Yan, Jianping Chen

**Affiliations:** 1College of Plant Protection, Shenyang Agricultural University, Shenyang 210095, China; meiq17@stu.syau.edu.cn; 2State Key Laboratory for Quality and Safety of Agro-products, Institute of Virology and Biotechnology, Zhejiang Academy of Agricultural Science, Hangzhou 310021, China; yangyong@zaas.ac.cn (Y.Y.); yesh@zaas.ac.cn (S.Y.); 15658000282@163.com (W.L.); xmwang@zaas.ac.cn (X.W.); zhoujie@zaas.ac.cn (J.Z.); yanchengqi@163.com (C.Y.); 3College of Plant Protection, Yunnan Agricultural University, Kunming 650000, China; 4Institute of Plant Virology, Ningbo University, Ningbo 315211, China; yuchulang@nbu.edu.cn

**Keywords:** H_2_O_2_, Rubisco activase, thylakoid membrane, chloroplast stoma, *Xanthomonas oryzae* pv. *oryzae*, *Oryza meyeriana*

## Abstract

*Oryza meyeriana* is a wild species of rice with high resistance to *Xanthomonas oryzae* pv. *oryzae* (*Xoo*), but the detailed resistance mechanism is unclear. Ribulose-1, 5-bisphosphate carboxylase/oxygenase (Rubisco) activase (RCA) is an important enzyme that regulates photosynthesis by activating Rubisco. We have previously reported that *Xoo* infection induced the relocation of RCA from the chloroplast stroma to the thylakoid membrane in *O. meyeriana*, but the underlying regulating mechanism and physiological significance of this association remains unknown. In this study, “H_2_O_2_ burst” with rapid and large increase in the amount of H_2_O_2_ was found to be induced by *Xoo* invasion in the leaves of *O.*
*meyeriana*. 3, 3-diaminobenzidine (DAB) and oxidative 2, 7-Dichlorodi-hydrofluorescein diacetate (H_2_DCFDA) staining experiments both showed that H_2_O_2_ was generated in the chloroplast of *O.*
*meyeriana*, and that this H_2_O_2_ generation as well as *Xoo* resistance of the wild rice were dramatically dependent on light. H_2_O_2_, methyl viologen with light, and the xanthine-xanthine oxidase system all induced RCA to associate with the thylakoid membrane in vitro, which showed that H_2_O_2_ could induce the relocation of RCA. In vitro experiments also showed that H_2_O_2_ induced changes in both the RCA and thylakoid membrane that were required for them to associate and that this association only occurred in *O. meyeriana* and not in the susceptible cultivated rice. These results suggest that the association of RCA with the thylakoid membrane helps to protect the thylakoid membrane against oxidative damage from H_2_O_2_. Therefore, in addition to its universal function of activating Rubisco, RCA appears to play a novel role in the resistance of *O. meyeriana* to *Xoo*.

## 1. Introduction

Rice (*Oryza sativa* L.) is a primary staple cereal and a source of food for more than half of the world’s population, but it is susceptible to many pathogens that can dramatically decrease yields. Rice bacterial blight is caused by *Xanthomonas oryzae* pv. *oryzae* (*Xoo*) and it is one of the most destructive diseases in all the major rice growing countries, typically reducing yields by 20% to 30%, but by as much as 50% in years when the disease is prevalent [1,2,3,4]. Chemical control of this disease is temporarily effective, but it often causes environmental pollution, and the most effective method of control is generally recognized to be through the use of resistant cultivars. Nevertheless, it is difficult for plant breeders to find new resistance genes from within the currently narrow genetic base of rice cultivars. Although they have low grain yields, wild species of the genus *Oryza* have adapted through natural selection to survive in harsh environments, and therefore often possess many useful traits that are not present in cultivated rice, including high disease resistance. Many wild species are highly resistant to bacterial blight, including *O. longistaminata* [5], *O. minuta* [6], *O. rufipogon* [7], and *O. officinalis* [8]. The resistance mechanisms of these wild rice species to *Xoo* have been studied in detail, and their resistance traits have been introduced into cultivated rice in breeding programs to give sustained improvement of rice productivity [5,6,7,8]. However, such resistance is often overcome by rapidly evolving *Xoo* populations and it is recognized that a better use of the genetic diversity of wild rice provides the best way to broaden the genetic base and adaptability of rice cultivars.

*O. meyeriana* is one of the wild relatives of rice growing in tropical and subtropical areas of South and Southeast Asia, and has long been known to possess a high resistance to *Xoo* pathogens: in 1981 and 1982 Peng et al. had reported that the *O. meyeriana* from Yunnan province, China, had high *Xoo* resistance [9,10]. Zhang et al. evaluated the *Xoo* resistance of 871 accessions of 13 wild rice species and found that the *O. meyeriana* accessions ranked as the most highly resistant [11]. Our previous studies also confirmed the high resistance to *Xoo* pathogens of this wild rice species [3,4,12,13,14,15]. However, there has been no research regarding the underlying mechanisms of this resistance, and this has limited its use in breeding programmes.

Pathogen invasion often leads to distinct inhibition of plant photosynthesis. In tomato, Berger et al. showed that infection by *Pseudomonas syringae* or *Botrytis cinerea* suppressed photosynthetic gene expression and downregulated the photosynthetic rate [16]. In *Arabidopsis*, Bonfig et al. found that invasion by *Pseudomonas syringae* decreased the chlorophyll fluorescence parameters (maximum PSII quantum yield, effective PSII quantum yield, and nonphotochemical quenching), which indicate a decline in the photosynthetic reaction [17]. We also found that *Xoo* inoculation decreased the photosynthetic rate in rice [4]. The effect of pathogen invasion on photosynthesis and the response of the plant photosynthetic apparatus to this invasion are extremely complex, due to the complicated structure and functional mechanism of photosystems [16,17,18]. On the one hand, the pathogen attempts to manipulate the photosynthetic metabolism of the plant for its own advantage. On the other hand, the plant has to reorganize carbon fluxes to provide resources to fight the pathogens. The ways in which these reactions interact with one another needs more research.

In green plants, Ribulose-1,5-bisphosphate carboxylase/oxygenase (Rubisco) catalyses the carboxylation of ribulose-1, 5-bisphosphate and CO_2_ to form two 3-phosphoglyceric acid molecules, and this is a dominant rate-controlling step in photosynthetic CO_2_ assimilation. Rubisco activase (RCA) is a key enzyme in the regulation of photosynthesis, and its primary role is to activate Rubisco for maintaining its catalytic competency. RCA encircles the Rubisco molecule and the two proteins are chemically cross-linked [19,20,21]. Through protein-protein interactions and ATP hydrolysis, RCA can remove sugar-phosphate inhibitors from Rubisco catalytic sites to allow for the binding of Mg^2+^ and ribulose 1,5-bisphosphate [20,22]. Numerous studies have shown that RCA is sensitive to inactivation that arises from various biotic [23,24] and abiotic [25,26,27] stresses. However, to our knowledge, no data are available at present concerning the response of RCA to *Xoo* invasion in the wild rice *O. meyeriana*.

We have reported that the inoculation of *O. meyeriana* with *Xoo* induced the relocation of RCA from the chloroplast stroma to associate with the thylakoid membrane [14]. This association simultaneously led to a decline in the initial activity and the activation status of Rubisco in the wild rice. However, the underlying regulatory mechanism and physiological significance of this binding to the thylakoid membrane is not known. In this study, we found that *Xoo* infection induced an “H_2_O_2_ burst” with a rapid and transient increase in the amount of H_2_O_2_ in the leaves of *O. meyeriana*. Experiments, both in situ and in vitro, revealed that a large amount of H_2_O_2_ was generated in the chloroplast and that this then induced the redistribution of RCA from the chloroplast stroma to the thylakoid membrane. In vitro experiments showed that H_2_O_2_ induced the changes in both RCA and the thylakoid membrane that are required for their association, and that this effect only occurred in the wild rice and not in cultivated rice. Based on these results, we suggest that the association of RCA with the thylakoid membrane helps to protect the thylakoid membrane against oxidative damage by H_2_O_2_. Thus, RCA has a novel role in the resistance of *O. meyeriana* to *Xoo* pathogens, in addition to its universal function of activating Rubisco.

## 2. Results

### 2.1. Resistance Analysis

The resistance of *O. sativa* cv. Dalixiang and *O. meyeriana* to *Xoo* was compared by measuring lesion length on leaves three weeks after inoculation with *Xoo* or H_2_O (as control). There was a highly significant difference in the resistance between the two hosts (*P* < 0.001): the *Xoo* lesions on Dalixiang were approximately 16.6 cm long, while those on *O. meyeriana* were only about 0.62 cm (Figure 1). These results are in agreement with the earlier observations that *O. meyeriana* has a high level of resistance to *Xoo* [3,11,12].

### 2.2. Effect of Xoo Inoculation on RCA Distribution

We first confirmed the earlier findings that *Xoo* infection induced the redistribution of RCA protein from the chloroplast stroma to the thylakoid membrane [14]. Intact chloroplasts were isolated from both *O. meyeriana* and the cultivar Dalixiang and the distribution of RCA between the chloroplast stroma and thylakoid membrane fractions were compared while using western blotting analysis. As shown in Figure 2, after mock inoculation with H_2_O, most RCA was found in the stroma fraction with only small amounts in the thylakoid membrane fraction of both plants. In cv. Dalixiang, there were no significant changes in either content or distribution of RCA after *Xoo* inoculation. However, in *O. meyeriana* RCA was sequestered to the thylakoid membrane from the stroma during the period from approximately 11 to 16 h after *Xoo* inoculation (Figure 2). The amounts of RCA in the soluble stroma decreased during this period and they did not recover to the original levels until approximately 17 h (Figure 2).

RCA activates Rubisco in the chloroplast stroma to maintain its catalytic competency, so the decline in the amount of RCA in chloroplast stroma decreased the initial activity and activation state of Rubisco in *O. meyeriana* not in *O. sativa* cv. Dalixiang [14]. The chlorophyll fluorescence parameters of Photosystem II (PSII) effective photochemical efficiency (*Φ_PSII_*) and PSII photochemical quenching (*qP*) reflect the part of PSII-absorbed light energy that was used for CO_2_ assimilation, and both of these parameters were notably decreased in *O. meyeriana* as the feedback of decline in CO_2_ assimilation due to the decreased Rubisco activation state (Appendix A). Plants have developed a photo-protective mechanism, which is called non-photochemical quenching (*qN*), to safely dissipate the excessive light energy as heat from PSII to protect the photosynthetic apparatus. In this study, in *O. meyeriana* inoculated with *Xoo*, *qN* was remarkably increased in parallel with the decrease in *Φ_PSII_* and *qP* (Appendix A). *F_v_/F_m_*_,_ PSII maximum photochemical efficiency, often reflects whether or not the photosynthetic apparatus has suffered irreversible damage. There was no significant change in *F_v_/F_m_* in *O. meyeriana* (Appendix A), which indicated that the photosystem of the wild rice was not destroyed by the excessive light energy that was caused by *Xoo* infection due to the effective working of the protective mechanism. There were no significant differences in *F_v_/F_m_*, *qP* and *qN* in Dalixiang in the period shortly after *Xoo* inoculation (Appendix A) when compared to the wild rice, but all of the three chlorophyll fluorescence parameters were dramatically decreased three weeks after inoculation (Appendix A), which indicated that the photosystem of Dalixiang was severely damaged. These results are in agreement with the difference in resistance between *O. meyeriana* and Dalixiang shown by the lesion lengths in Figure 1.

There was no significant decline in the amount of RCA in the thylakoid membrane fraction of infected *O. meyeriana* following incubation in the HEPES-KOH buffer, and RCA was hardly detected in the buffer until the incubation lasted for 24 h (Figure 3A). The thylakoid membrane was also washed with 2 M NaCl or 2 M NaBr, or treated for 30 min. with 1% Triton X-100. Washing with 2 M NaBr has been shown to remove extrinsic thylakoid proteins [28]. However, none of these treatments released noticeable amounts of RCA (Figure 3B). However, limited proteolysis with thermolysin did decrease the amount of RCA in the thylakoid membrane (Figure 3B), indicating that thermolysin digested RCA thorough its protease activity. All of these results suggest that, in the wild rice, *Xoo* infection induces a tight association of RCA with the thylakoid membrane.

### 2.3. Effect of Xoo Inoculation on H_2_O_2_ Generation

H_2_O_2_ is a central signaling molecule that induces plant responses to pathogens. The H_2_O_2_ content of leaves was measured at different times after inoculation with *Xoo* to study the effect of *Xoo* inoculation on H_2_O_2_ generation. In leaves of cv. Dalixiang, H_2_O_2_ content increased slightly during the period 0-16 h after *Xoo* inoculation, and then remained stable (Figure 2). In *O. meyeriana* leaves inoculated with *Xoo*, there was a rapid increase in H_2_O_2_ content after approximately 10 h, peaking at about 13 h, and then declining rapidly until 16 h, after which the decrease was slight (Figure 2). During this period, the H_2_O_2_ content in the wild rice was much greater than in the cultivated rice; for example, at 13 h after *Xoo* inoculation, the H_2_O_2_ content was 12.70 µM g^−1^ FW in *O. meyeriana,* but only 4.49 µM g^−1^ FW in Dalixiang (Figure 2).

It is noteworthy that the period of rapid increase in H_2_O_2_ content was remarkably co-incident with that for the association of RCA with the thylakoid membrane (Figure 2), which strongly suggests some relationship between the rapidly increasing H_2_O_2_ content and the relocation of RCA within the chloroplasts of *O. meyeriana*.

### 2.4. Subcellular Localization of H_2_O_2_

We next investigated the subcellular localization of *Xoo*-induced H_2_O_2_ to determine whether it might be in contact with RCA. Two different methods were used.

After 3, 3-diaminobenzidine (DAB) staining, the mesophyll cells of *O. meyeriana* were stained brightly brown 13 h after *Xoo* inoculation as compared to the mock (water-inoculated) controls (Figure 4B), which indicates that *Xoo* infection induced a large H_2_O_2_ accumulation in the leaves of the resistant wild rice. This accumulation was mostly in the chloroplasts (Figure 4B). However, H_2_O_2_ accumulation was significantly less at 24 h after inoculation (Figure 4B). In contrast, there was little H_2_O_2_ accumulation in the leaves of Dalixiang at either 13 or 24 h after inoculation (Figure 4A). Therefore, these results are consistent with those from H_2_O_2_ examination of leaf extracts, as described above (Figure 2). In the water-inoculated control leaves, the color from DAB staining was slightly deeper in *O. meyeriana* than in Dalixiang, perhaps because the basal H_2_O_2_ content of the wild rice was slightly higher than that of the cultivated rice.

After oxidative 2, 7-Dichlorodi-hydrofluorescein diacetate (H_2_DCFDA) staining, there was no fluorescence of mesophyll cells of the cultivated rice that had been inoculated with either water or *Xoo* (Figure 5A). Green fluorescence was also hardly found in the wild rice that was inoculated with water (Figure 5B). However, strong fluorescence was detected in the wild rice at 13 h after *Xoo* inoculation, although this was much weaker by 24 h (Figure 5B). This fluorescence also appeared to overlap with the auto-fluorescence of the chloroplasts (Figure 5B). These results all show that the H_2_O_2_ induced by *Xoo*-infection mainly accumulates in the chloroplast, which is in agreement with the results from DAB staining.

### 2.5. Light-Dependent H_2_O_2_ Generation and Resistance in Wild Rice

We next studied the effects of shading on H_2_O_2_ concentrations and *Xoo* resistance (Figure 6A) to further investigate the relationship between chloroplast H_2_O_2_ and *Xoo* resistance in the wild rice. Although *Xoo* inoculation induced a large accumulation of H_2_O_2_ in chloroplasts under light conditions, there was little accumulation without light (Figure 6C), which indicated that H_2_O_2_ accumulation in the chloroplasts was strongly light dependent. In the presence of light, the wild rice was also highly resistant to *Xoo* and lesions were very short, but in the absence of light, the plants became much more susceptible (Figure 6B). These results suggest that, in the wild rice, H_2_O_2_ in the chloroplasts stimulates the resistance to *Xoo*.

### 2.6. Effect of H_2_O_2_ on RCA Redistribution in Vitro

Different methods were then used to generate H_2_O_2_ in vitro and to test whether this H_2_O_2_ could induce the redistribution of RCA from the soluble stroma to the thylakoid membrane. The first method used methyl viologen (MV), which, in the light, can transfer electrons from Photosynthetic system I (PSI) to molecular H_2_O to produce H_2_O_2_ in plant chloroplasts [29,30]. The results showed that MV induced the moving of RCA from the stroma to the thylakoid membrane in the resistant wild rice, but not in the sensitive cultivated rice, and that this effect was light dependent (Figure 7). H_2_O_2_ induced RCA association with thylakoid membrane with or without light in *O. meyeriana* but not in Dalixiang (Figure 7). Xanthine (XAN)-xanthine oxidase (XOD) is a kind of oxidative system that can produce H_2_O_2_ in vitro [31,32,33]. When used alone, neither XAN nor XOD induced the association of RCA with the thylakoid membrane in either rice species, but when used together, RCA became bound to the thylakoid membrane in *O. meyeriana,* but not in Dalixiang, whether treated with light or not (Figure 7). All of these results show that H_2_O_2_ induces the redistribution of RCA to the thylakoid membrane from the soluble stroma in the resistant rice *O. meyeriana*.

### 2.7. In Vitro Investigation of the Association between RCA and Thylakoid Membrane

In vitro experiments were then performed to investigate factors that might affect the association of RCA with the thylakoid membrane in *O. meyeriana*. In experiments where H_2_O_2_ was used for different lengths of time, RCA began to redistribute from the stroma to thylakoid membrane after approximately 10 min. of H_2_O_2_ treatment. The amounts of RCA in the thylakoid membrane increased greatly after approximately 30 min. treatment, and then seemingly reached a plateau that was not exceeded by longer treatments (Figure 8A).

Next, the stroma and thylakoid membrane fractions were separately treated with H_2_O_2_ for 30 min. before mixing and incubation. RCA redistribution occurred within 5 min. of mixing (Figure 8B), which suggested that the 30 min. pre-treatment might have changed the RCA structure ready for association with the thylakoid membrane. As expected, a large amount of RCA was detected in the thylakoid membrane (Figure 8B). When the stroma fraction was pre-treated with H_2_O_2_ for 30 min. and it was then added to an untreated thylakoid membrane fraction, there was no redistribution of RCA after 5 min. (Figure 8B). Large amounts of RCA were detected in the thylakoid membrane fraction after 45 min. since the thylakoid membrane was exposed to H_2_O_2_ after mixing (Figure 8B). Pre-treatment of the thylakoid membrane fraction before mixing it with an untreated stroma fraction similarly resulted in little redistribution of RCA after 5 min. (Figure 8B). These results show that changes in the thylakoid membrane and in RCA take place to facilitate their H_2_O_2_-induced association.

In further experiments, the thylakoid membrane fraction that was isolated from *O. meyeriana* and the stroma fraction isolated from Dalixiang were separately treated with H_2_O_2_ before mixing and incubation. The reciprocal experiment was also done using the thylakoid membrane fraction from Dalixiang and the stroma fraction from *O. meyeriana.* In neither case did RCA redistribute to the thylakoid membrane (Figure 8C), showing that the redistribution effect was specific to both the RCA and the thylakoid membrane of the resistant wild rice and not to the susceptible cultivated rice.

## 3. Discussion

### 3.1. High Resistance of O. meyeriana to Xoo

Wild species of rice have adapted through natural selection to survive in harsh environments and they possess many useful traits that are not present in cultivated rice. For example, *O. nivara* has high resistance to the rice brown planthopper [34], *O. officinalis* is highly resistant to the rice small brown planthopper [35], and *O. longistaminata* and *O. rufipogon* are reported to have resistance to *Xoo* [5,7]. *O. meyeriana* is one of the wild relatives of rice growing in tropical and subtropical areas of South and Southeast Asia. In China, it is distributed in the Yunnan and Hainan provinces. This wild rice has been shown to possess a high resistance to *Xoo* pathogens [11] and this was further confirmed here (Figure 1). We have previously introduced this trait into a rice cultivar while using asymmetric somatic hybridization [3,12], but studies of the mechanism of this resistance are needed if this is to be widely used for genetic improvement of rice.

### 3.2. H_2_O_2_ Generation in Chloroplasts of O. meyeriana Infected by Xoo

H_2_O_2_ is recognized to be an important signaling molecule that triggers various downstream resistance reactions in the response of plants to pathogen infection [36,37,38,39]. In many plant resistance reactions, one of the earliest cellular responses following successful pathogen recognition is a large and transient production of reactive oxygen species, mainly H_2_O_2_, a phenomenon that is often called an “oxidative burst” [38,39,40,41,42]. In the present study, the inoculation of *O. meyeriana* with the bacterial pathogen *Xoo* also triggered an “oxidative burst” (Figure 2).

Plasma membrane-bound NADPH/NADH oxidase or cell wall-localized peroxidase are known to be the major sources of H_2_O_2_ during the “oxidative burst” [42,43]. NADPH/NADH oxidase, known as the respiratory burst oxidase, catalyzes the production of superoxide by one-electron reduction of molecular oxygen while using NADPH as an electron donor, and then other enzymes, such as superoxide dismutase catalyze superoxide, to its more stable dismutation product H_2_O_2_ [44]. The peroxidase catalyzes the oxidoreduction of various substrates to generate H_2_O_2_ [44,45]. In the “oxidative burst”, NADPH/NADH oxidase is localized in the plant plasma membrane and peroxidase is localized in the cell wall, and H_2_O_2_ generation is initiated in apoplasts and the extracellular spaces of plant tissues.

In addition to NADPH/NADH oxidase and peroxidase, the electron transport chain in the chloroplast is another important source of H_2_O_2_ in plants. However, it is not known whether and how H_2_O_2_ generation in chloroplasts is involved in plant resistance to pathogens. In this study, the results from both DAB staining and H_2_DCFDA fluorescence imaging showed that H_2_O_2_ was largely and transiently generated in the chloroplasts of the resistant wild rice, but not in the susceptible cultivated rice, when it was inoculated with *Xoo* (Figure 4 and Figure 5). Additionally, both *Xoo* resistance and H_2_O_2_ generation in *O. meyeriana* were distinctly light-dependent (Figure 6). Therefore, these results provide evidence that H_2_O_2_ from chloroplasts triggered the high *Xoo* resistance of *O. meyeriana*. Liu et al. and Caplan et al. have also demonstrated that chloroplasts are one of the major sources of the “oxidative burst” mediating the resistance of tobacco plants to tobacco mosaic virus [40,41]. Thus, the “oxidative burst” might be initiated as a defense response in plant chloroplasts as well as the apoplast.

Light is the important energy source that plants use to regulate their growth and development via various energy-dependent reactions in addition to the reaction of photosynthesis. In chloroplasts, light energy beyond what is used for photosynthesis can be used to generate reactive oxygen species, mainly H_2_O_2_, by a series of complicated oxidoreductive systems, and the regulation of the redox state in chloroplasts has been widely proven to play an important role in plant response to different stresses, such as cold [46], drought [47], salt [48,49], and waterlogging [50]. The “oxidative burst” is a process that is not only requires energy, but is also fine-regulated in the redox state. Therefore, it is entirely plausible that the H_2_O_2_ used by *O. meyeriana* in *Xoo* resistance is generated in the chloroplast.

### 3.3. Chloroplast H_2_O_2_ Induced the Association of RCA with Thylakoid Membrane Following Infection of O. meyeriana with Xoo

RCA is an important enzyme regulating photosynthesis. An increasing number of studies have been made regarding the structure and function of this enzyme since it was discovered based on an *rca* mutant of *Arabidopsis* in the 1980s. However, there are only a few reports on the cellular localization of RCA. Anderson and Carol found that labeling for RCA occurred in the chloroplast of pea plants [51]. Hong et al. also demonstrated that the RCA protein was distributed in the chloroplast of bundle sheath and mesophyll cells of the C_4_ plant *Amaranthus tricolor* [52]. Immuno-gold localization by Jin et al. showed that RCA was mainly detected in the chloroplast stroma and to a smaller extent in the thylakoid membrane in rice [53]. Interestingly, Chen et al. had demonstrated that the amount of RCA in the thylakoid membrane increased when the pH gradient across the thylakoid membrane and ATP in the chloroplast decreased in cultivated rice [54]. It has also been reported that moderate heat treatment (42 ℃) induced the redistribution of RCA from the stroma to the thylakoid membrane in spinach and tobacco [55,56]. We also reported that *Xoo* inoculation induced the association of RCA with the thylakoid membrane in the wild rice *O. meyeriana* [14] Figure 2. However, the mechanism that underlies the redistribution of RCA within the chloroplasts of the wild rice still remained unstudied.

A significant finding of the present study was that the period when *Xoo* infection triggered the “H_2_O_2_ burst” precisely coincided with the movement of RCA from the stroma to the thylakoid membrane (Figure 2), which suggested a causal link. In vitro experiments showed that H_2_O_2_ induced the redistribution of RCA in the wild rice but not in the cultivated rice, and that the effect could be reproduced if H_2_O_2_ was generated by two other methods, using MV-light or XAN-XOD. Additionally, the newly-generated H_2_O_2_ triggered by *Xoo* was clearly localized in chloroplasts (Figure 4 and Figure 5), and the resistance of *O. meyeriana* to *Xoo* and the H_2_O_2_ accumulation were both dependent on light. Taken together, we therefore propose that the chloroplast-generated “H_2_O_2_ burst” triggered by *Xoo* infection induces the association of RCA with the thylakoid membrane.

H_2_O_2_ is a small oxidative molecule that can cause significant modifications in the protein structure in the redox state and that is known to trigger a series of signaling transduction steps in response to various biotic or abiotic stresses in plants [18,38,39,40,42,47]. It seemed that H_2_O_2_ might induce a structural change in RCA to make it suitable for binding to the thylakoid membrane. Further experiments revealed that some H_2_O_2_-induced changes in the thylakoid membrane were also required for this association (Figure 8). Moreover, this H_2_O_2_-induced association was specific to the RCA and thylakoid membranes of *O. meyeriana,* rather than to those of the cultivated rice (Figure 8). It therefore appears that many changes in both RCA and the thylakoid membrane are required for them to associate with each other. Further study is needed to understand the detailed mechanism.

### 3.4. Potential Novel Role of RCA in the Xoo Resistance of O. meyeriana

It has been well documented that the primary role of RCA is to activate Rubisco to maintain its catalytic competency, and it is therefore expected that RCA would be localized in the chloroplast stroma coupled with Rubisco. Any RCA bound to the thylakoid membrane would not be involved in activating Rubisco, because the membrane is not the place where carboxylation takes place. Therefore, it seems likely that RCA has an additional role in the thylakoid membrane. Rokka et al. and Yang et al. found an increase in the amount of RCA that was bound to the thylakoid membrane under moderate heat treatment (42 ℃) and suggested that this was part of a system that helped to protect the thylakoid-associated translation machinery against heat inactivation [55,56]. However, Chen et al. proposed that the association of RCA with the thylakoid membrane was involved in the RCA activation of Rubisco due to its dependence on ATP and the pH gradient across the thylakoid membrane [54]. In the present study, the association of RCA with the thylakoid membrane was not found in the cultivated rice, but only in the wild rice, which suggested a resistance role rather than a general physiological one.

Reactive oxygen species damage plant cells because of their strong oxidative toxicity to membranes, proteins, pigments, and other essential macro-molecules, and such damage has been widely reported in plants under different biotic or abiotic stresses [47,48,57,58,59]. Therefore, H_2_O_2_ generated in the “oxidative burst” in response to pathogens could be double-edged: on the one hand it acts as a signaling molecule triggering the response reaction, but on the other hand it damages plant cells as an unstable oxidative molecule. The thylakoid membrane is crucial for the photosynthetic reaction as the site of various proteins and pigments that are involved in photosynthetic electron transport and photophosphorylation. The rapid binding of RCA to the thylakoid membrane might help to protect the thylakoid membrane system against the oxidative damage that might otherwise occur from the locally-produced H_2_O_2_ burst.

RCA is related to an AAA^+^ family of proteins, a class of chaperone-like ATPases associated with a variety of cellular activities, including the assembly, operation, or disassembly of protein complexes [60]. Proteins in the AAA^+^ family are also often covalently linked to other protein domains that mediate localization to cellular membranes. Therefore, RCA bound to the membrane could play a role as a chaperone in protecting the thylakoid-associated proteins against H_2_O_2_ damage. This protection might be helpful in explaining the high resistance of *O. meyeriana* to *Xoo* infection (Figure 1). On the basis of the results in this study, we therefore suggest that RCA may be involved in the resistance of *O. meyeriana* to *Xoo* in addition to its universal function in activating Rubisco.

*O. meyeriana* is highly resistant to *Xoo*, perhaps as the result of many mechanisms, rather than a single one. For example, we have found that the xylem secondary cell-wall thickening is involved in this resistance [4]. Here, we report that RCA is one of the factors that contribute to the resistance of *O. meyeriana*. Additionally, we have transferred the trait of this resistance of *O. meyeriana* into cultivated rice while using asymmetric somatic hybridization. Using the hybrid progeny as a resistant parent, we constructed a segregating population and mapped three major quantitative trait loci (QTLs) for *Xoo* resistance on chromosomes 1, 3, and 5 [3]. It seems that the three major QTLs are not associated with the *RCA* gene, because they are at different loci of different chromosomes, indicating that the *RCA* gene from *O. meyeriana* was not transferred into the cultivated rice through asymmetric somatic hybridization. However, the *RCA* gene of the wild rice might interact with the major QTLs to make a contribution to the high resistance, so it is necessary that these QTLs are isolated and characterized by map-based cloning and that additional genetic analysis is conducted to reveal the relationship between the *RCA* gene and the three major QTLs for resistance.

## 4. Materials and Methods 

### 4.1. Plant Materials

*O. meyeriana*, a wild species of rice with high *Xoo* resistance, was from the Zhejiang Academy of Agricultural Sciences collection. It was originally provided by Professor Xin-hua Wei, China National Rice Research Institute and its accession No. was ABB658. *O. sativa* ssp. *japonica* (cv. Dalixiang), a rice cultivar that is susceptible to rice *Xoo* pathogens, was from the Zhejiang Academy of Agricultural Sciences collection. The rice plants were grown at 28/25 °C (day/night; 16 h/8 h), with photon flux density 600–800 µM m^−2^ s^−1^, and relative humidity 60–80%.

### 4.2. Pathogen Inoculation and Resistance Identification

In this study, *Xoo* strain PXO99 was used to inoculate the rice leaves. The leaf-clipping method was used to inoculate *Xoo* pathogens, as previously described [61,62]. Briefly, The *Xoo* bacterial inoculum was prepared from a 48 h culture on potato semisynthetic agar (PSA) [62] and its density was adjusted to 10^9^ CFU/mL. Approximately 40 days after transplanting (booting stage), a pair of scissors was dipped into the inoculum and used to clip about 1 cm from the tip of the fully expanded leaves. At least, 16 leaves from four plants of each of the wild and cultivated rice were inoculated with the bacteria. Bacterial lesion length was measured for resistance evaluation after three weeks of inoculation.

### 4.3. Measurements of Chlorophyll Fluorescence

Chlorophyll fluorescence was measured with an integrating chlorophyll fluorometer (LI-6400 leaf chamber fluorometer, LI-COR, USA). The rice leaves were kept in dark for 1 h before the minimum fluorescence yield in the dark (*F_o_*) was recorded under a measuring light (0.1 µmol m^−2^ s^−1^). The maximum fluorescence yield in the dark (*F_m_*) was then measured with a 600 ms saturating flash (8000 µmol m^−2^ s^−1^). Subsequently, an actinic light (1000 µmol m^−2^ s^−1^*)* was applied for 30 min. to obtain the stable fluorescence yield during actinic light (*F_s_*), which was followed by a measurement of the maximum fluorescence yield druing actinic illumination (*F_m_’*) with another saturating flash. A far-red light was then applied to determine the minimum fluorescence yield (*F_o_’*). Different chlorophyll fluorescence parameters were defined, as follows:


*F_v_/F_m_ = *(*F_m_ − F_o_*)*/F_m;_*



*Φ_PSII_ = *(*F_m_’ − F_s_*)*/F_m_’;*



*qP = *(*F_m_’ − F_s_*)*/*(*F_m_’ − F_o_’*)*;*



*qN = [*(*F_m_ − F_m_’*)* − *(*F_o_ − F_o_’*)*]/*(*F_m_ − F_o_*)*.*


### 4.4. Isolation and Sub-Fractionation of Chloroplasts

Intact chloroplasts were isolated according to Cline et al. and Gunasinghe and Berger with some modifications [63,64]. In the procedure finally adopted, leaves (20 g) were homogenized in 200 mL of grinding buffer containing 330 mM sorbitol, 50 mM Hepes-KOH (pH 7.5), 2 mM EDTA, 1 mM MgCl_2_, 1 mM MnCl_2_, 0.5% BSA, and 4 mM ascorbic acid until the tissue was smaller than 1 mm^2^. The homogenate was filtered through eight layers of cheesecloth, and centrifuged for 1 min. at 4000 rpm in a Sorvall GSA rotor at 4 ℃. The pellet was suspended in 8 mL of grinding buffer and then overlayered onto a 40% to 80% Percoll gradient containing 0.6 mM glutathione, gradients of 0.7% to 2.7% (w/v) of polyethyleneglycol 3350 and 0.25% to 0.92% (w/v) of both BSA and Ficoll 400, and the same ingredients as in the grinding buffer. The gradient tubes were centrifuged for 8 min. at 7500 rpm in the Sorvall HB-4 rotor. Two green-colored bands were found in the tube after centrifugation. The lower green band of intact chloroplasts was collected and diluted with about 25 mL of grinding buffer, and then was centrifuged for 5 min. at 3500 rpm in the HB-4 rotor. The pellet was suspended and centrifuged again in 25 mL of HSM suspension medium containing 50 mM Hepes-KOH (pH 8.0), 330 mM sorbitol, and 42 mM methionine. The pellet was washed again with HSM and drained at 4 ℃. The resulting pellet of intact plastids was resuspended in about 3 mL HSM and the final concentration was adjusted to 3.0 mg Chl/mL. The intactness of the chloroplasts was examined by observation under an inverted microscope (ECLIPSE Ts2-FL, Nikon, Japan). If there were ruptured chloroplasts in the HSM buffer, the buffer would be re-extracted by repeating the procedure until the intact chloroplasts remained. 

Purified intact chloroplasts were ruptured in 1 mM PMSF and the suspension was placed on ice for 45–60 min. then at room temperature for 30 min. The suspension of broken chloroplasts was centrifuged for 10 min. at 14000 g. The sediment was dissolved in loading buffer as the membrane fraction for SDS-PAGE analysis. An equal volume of precooled 20% TCA was added to the supernatant and the mixture was incubated for at least 30 min. on ice, followed by centrifugation for 20 min. at 14000 g. The stroma proteins were washed twice with 80% acetone and drained at 4 ℃, and were then dissolved in loading buffer for SDS-PAGE analysis.

### 4.5. SDS-PAGE and Immunological Analyses

Two fractions of thylakoid membrane and stroma of chloroplasts were separated by differential centrifugation following an osmotic shock to intact chloroplasts. SDS-PAGE with 12% (w/v) acrylamide gels was used to separate the proteins in the fractions and the samples were loaded on the total protein basis (5 µg protein). Antibodies that were raised against RCA were used to detect the RCA proteins in western blotting after the polypeptides had been electrophoretically transferred to a polyvinylidene fluoride (PVDF) membrane (Bio-Rad, Hercules, USA). RCA antibodies were produced while using prokaryotically expressed products as antigens of *RCA* genes that were derived from *O. meyeriana* and the cultivar rice, respectively. After the proteins were transferred to the PVDF membrane, the secondary anti-mouse IgG antibody conjugated with alkaline phosphatase (Sigma) and the nitroblue tetrazolium (NBT)/5-bromo-4-chloro-3-indolyl phosphate (BCIP) system were used to detect and visualize the immunoblotting signals. 

### 4.6. Determination of H_2_O_2_ in Leaf extracts

The content of H_2_O_2_ in leaves was measured by examining the absorbance of the titanium-peroxide complex at 415 nm, according to a previous method [65]. One g fresh leaves was homogenized in an ice bath with 10 mL 0.1% TCA. After the homogenate was centrifuged at 10,000 *g* for 15 min., one mL of the supernatant was added to 1 mL of 10 mM potassium phosphate buffer with 2 mL of 1M KI and pH 7.0, and the absorbance of the supernatant was then measured at 415 nm. The absorbance was quantified while using a standard curve generated from known concentrations of H_2_O_2_.

### 4.7. Subcellular Localization of H_2_O_2_

Two different approaches were used on rice leaf frozen sections to visualize the subcellular localization of H_2_O_2_ in leaves. DAB staining was performed, as previously described [66,67]. Briefly, DAB (Sigma) was dissolved in 10 mM Tris-acetate (pH 5.0) at a concentration of 1 mg mL^−1^. Rice leaf sections were incubated in DBA solution for 3 h at room temperature and then were washed three times with distilled water. Subsequently, the sections were observed and photographed with an inverted microscope (ECLIPSE Ts2-FL, Nikon, Japan) at 200× magnification. A second method with H_2_DCFDA fluorescence imaging was conducted while using an H_2_O_2_ Assay Kit (Invitrogen), according to the manufacturer’s instruction. A 10 mM stock solution of H_2_DCFDA (Molecular Probes) was prepared by dissolving H_2_DCFDA in anhydrous dimethyl sulfoxide, and after a subsequent dilution step with distilled water, a 10 µmol working concentration was set. The rice leaf sections were incubated in the working solution for 30 min., and then washed three times with distilled water. Green fluorescence from oxidized H_2_DCFDA was monitored by confocal laser scanning microscopy (LSM510, Zeiss, Germany) while using the dye-specific excitation/emission wavelength of 488/529 nm. Background chloroplast autofluorescence emission was monitored with a 650 nm long-pass filter.

### 4.8. Salt Washing and Thermolysin Treatment of Isolated Thylakoids

The thylakoid membrane fractions with 2 mg chlorophyll mL^−1^ were washed with 2 M NaCl or 2 M NaBr in 10 mM HEPES-KOH with 5 mM MgCl_2_ (pH 7.8), and then incubated in 1% (v/v) Triton X-100 for 30 min. on ice. The thylakoid membrane fractions were then added to in 0.4 mg mL^−1^ of thermolysin (pH 7.8, Sigma, St. Louis, USA) with 50 mM Tricine, 100 mM sorbitol, 5 mM NaCl, 10 mM MgCl_2_, and 2 mM CaCl_2_ for about 8 min. on ice, after which EDTA (50 mM) was added to stop the digestion. The thylakoid membrane fractions were then collected by 10 min. centrifugation at full speed in an Eppendorf centrifuge. 

### 4.9. Oxidative Treatment of Ruptured Chloroplasts

A suspension of ruptured chloroplasts was mixed with 10 mM of H_2_O_2_ (Sigma, St. Louis, USA), 20 µM MV (Sigma, St. Louis, USA), 10 mM XAN (Sigma, St. Louis, USA), or 10 mM XOD (Sigma, St. Louis, USA), and the mixture was then incubated for 30 min. at room temperature usually using a light photon flux density of about 600 µM m^−2^ s^−1^, but up to 1000 µM m^−2^ s^−1^ for the MV treatment to stimulate enough H_2_O_2_ production from the photosynthetic electron transport chain. The chloroplasts were then divided into thylakoid membrane and stroma fractions for RCA protein detection, as described above.

### 4.10. Shading Treatment of Wild Rice Leaves

To investigate the influence of light on the resistance of the wild rice to *Xoo*, the leaves were shaded with a 2–3 cm piece of aluminum foil for 1 wk under the light condition of 600 µM m^−2^ s^−1^ after inoculation. The foil was then removed and light was re-instated for 4 d. Subsequently, the lengths of lesions on inoculated leaves were measured for resistance evaluation, and the accumulation and subcellular localization of H_2_O_2_ was determined.

## Figures and Tables

**Figure 1 plants-08-00351-f001:**
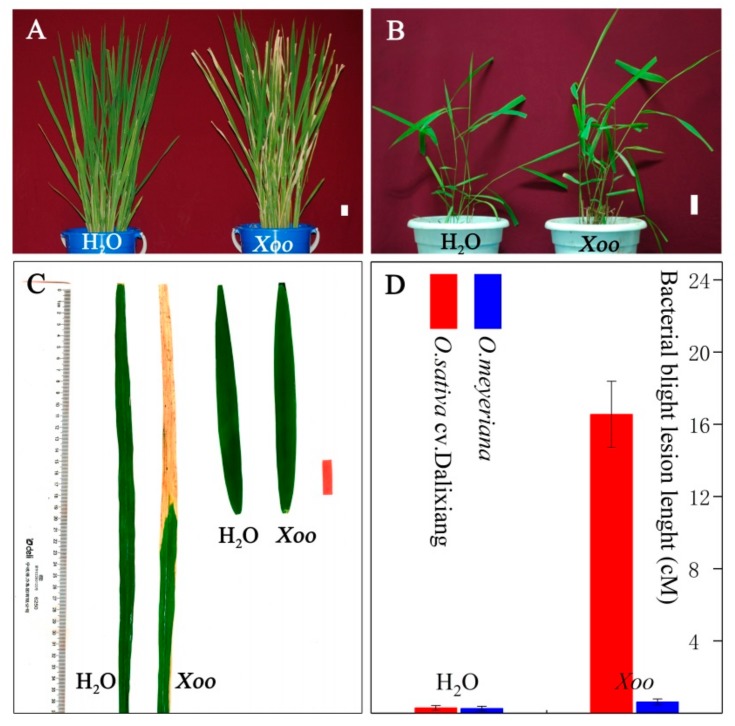
The high resistance of *Oryza meyeriana* to *Xanthomonas oryzae* pv. *oryzae* (*Xoo*). Rice was inoculated by the leaf-clipping method and at least 16 leaves from four plants were inoculated. Lesion length was measured three weeks after inoculation. (**A**) *O. sativa* ssp. *japonica* cv. Dalixiang. Left is plant inoculated with water (control) and right is plant inoculated with *Xoo*. White scale bar = 3 cm. (**B**) *O. meyeriana*. Left is plant inoculated with water and right is plant inoculated with *Xoo*. (**C**) Leaves inoculated with water and *Xoo*. Left is Dalixiang and right is *O. meyeriana*. Red scale bar = 3 cm. (**D**) Lesion length caused by *Xoo* infection. Bars indicate the standard error.

**Figure 2 plants-08-00351-f002:**
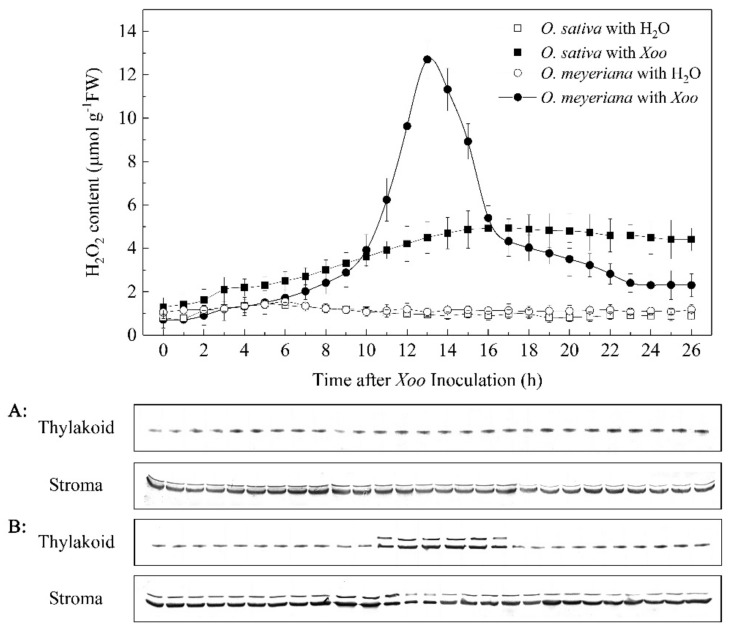
Effect of *Xoo* inoculation on H_2_O_2_ content in the leaves of Dalixiang and *O. meyeriana* during a 26 h test period (upper panel) and rubisco activase (RCA) content in the thylakoid membrane fraction and soluble stroma fraction of chloroplasts in Dalixiang and *O. meyeriana* (lower panel). (**A**): Dalixiang, (**B**): *O. meyeriana*. Intact chloroplasts were isolated from rice leaves at different times after *Xoo* inoculation. The protein samples were loaded on the basis of 5 µg total protein and western blot analysis was performed with an antibody raised against RCA.

**Figure 3 plants-08-00351-f003:**
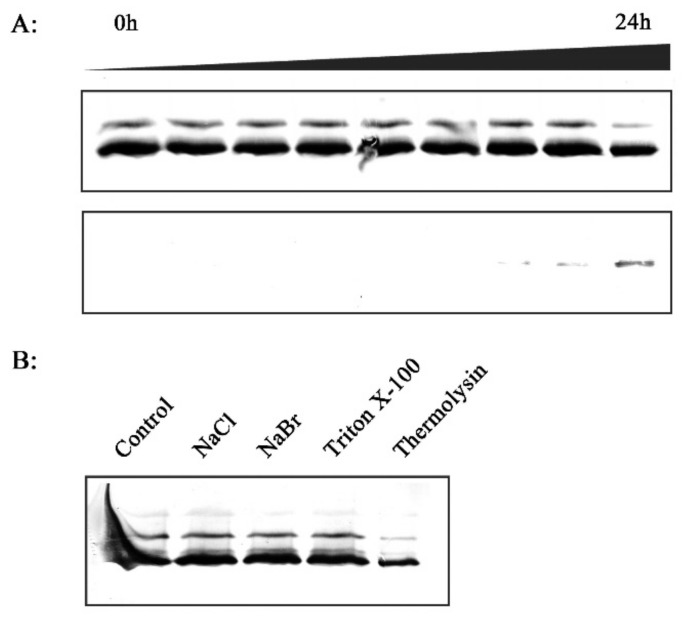
Stability of the association of RCA with the thylakoid membrane. (**A**) Thylakoids were isolated from leaves of *O. meyeriana* inoculated with *Xoo* and then incubated in HEPES-KOH buffer solution to release RCA from the thylakoid membrane for 0, 1/6, 1/2, 1, 2, 3, 6, 12, 24 hours. (**B**) Isolated thylakoid membrane washed with 2 M NaCl or 2 M NaBr, or treated for 30 min. with 1% Triton X-100.

**Figure 4 plants-08-00351-f004:**
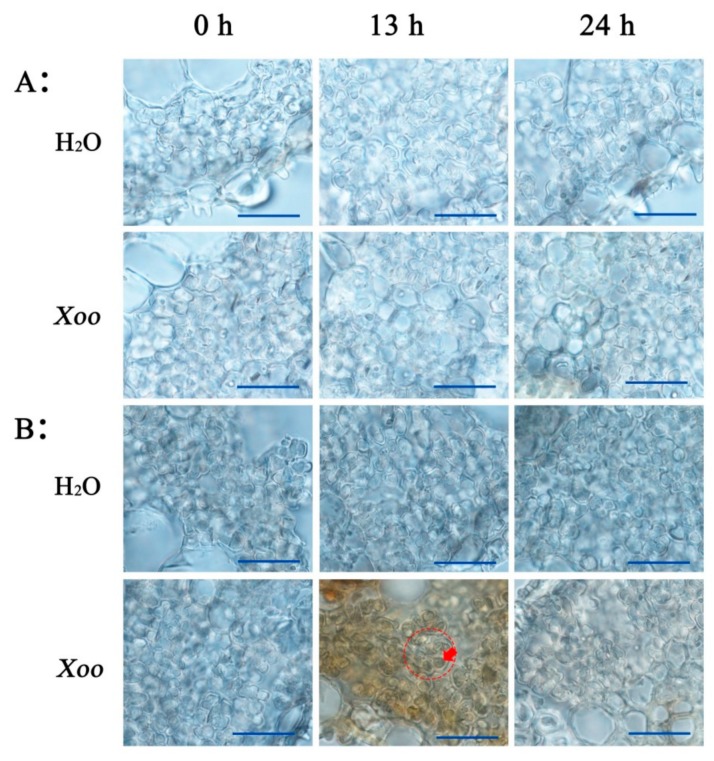
Subcellular localization of H_2_O_2_ detected by After 3, 3-diaminobenzidine (DAB) staining in Dalixiang and *O. meyeriana* 0 h, 13 h, and 24 h after *Xoo* inoculation. Rice leaf sections were incubated in 3, 3-diaminobenzidine (DAB) solution and then examined in an inverted microscope. (**A**): Dalixiang, (**B**): *O. meyeriana*. Red circle indicates a mesophyll cell and the arrow indicates a chloroplast. Scale bars = 20 µm.

**Figure 5 plants-08-00351-f005:**
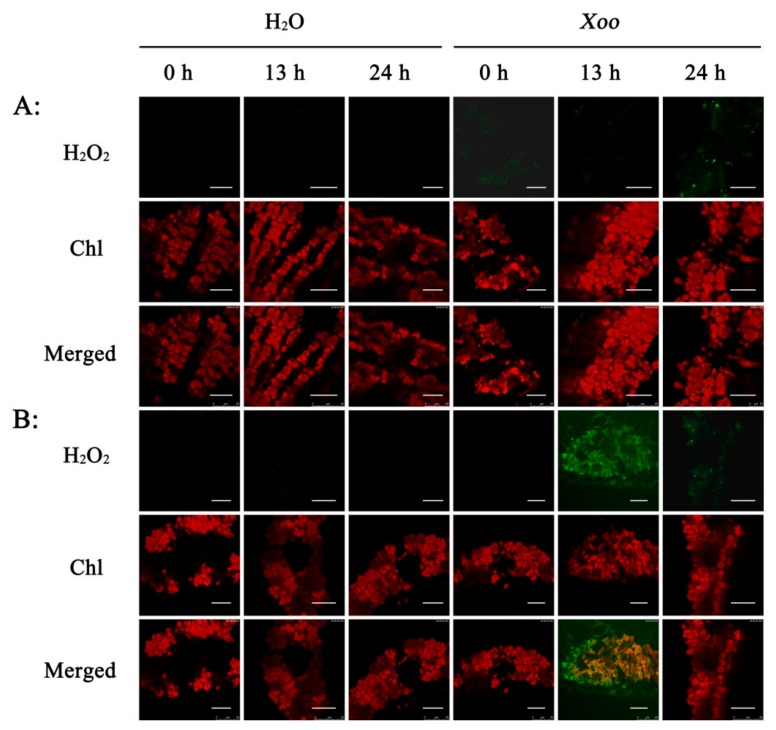
Subcellular localization of H_2_O_2_ detected by H_2_DCFDA fluorescence imaging in Dalixiang and *O. meyeriana* 0 h, 13 h, and 24 h after *Xoo* inoculation. Rice leaf sections were incubated in 2, 7-Dichlorodi-hydrofluorescein diacetate (H_2_DCFDA) solution and then examined by confocal laser scanning microscopy with excitation/emission wavelength of 488/529 nm. Chloroplast autofluorescence emission was 650 nm. (**A**): Dalixiang, (**B**): *O. meyeriana*. Chl: chloroplast. Scale bars = 20 µm.

**Figure 6 plants-08-00351-f006:**
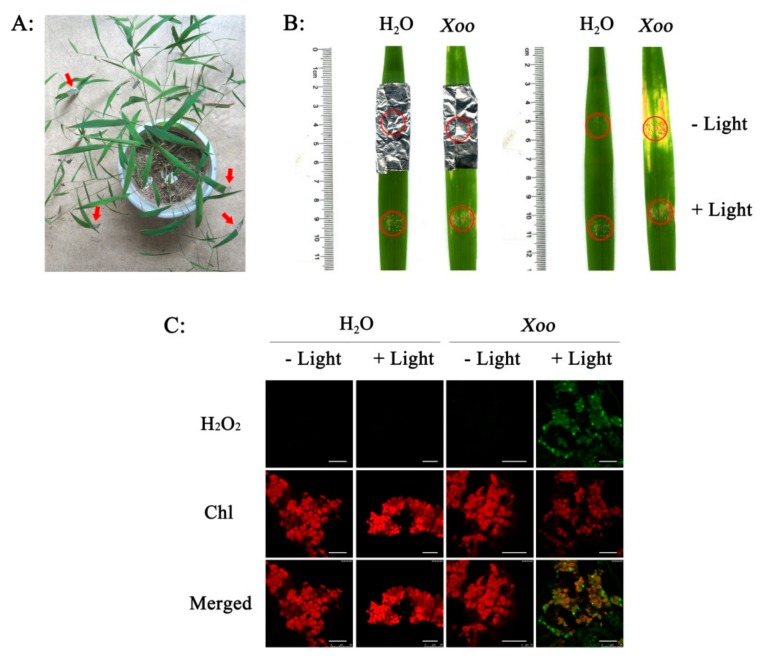
Resistance to Xoo and H_2_O_2_ accumulation in chloroplasts were dependent on light in *O. meyeriana*. (**A**) Shading treatment with aluminum foil to the leaves of *O. meyeriana*. (**B**) Effect of light on the lesion length caused by Xoo infection. (**C**) Effect of light on the subcellular localization of H_2_O_2_. Chl: chloroplast. Scale bars = 20 µm.

**Figure 7 plants-08-00351-f007:**
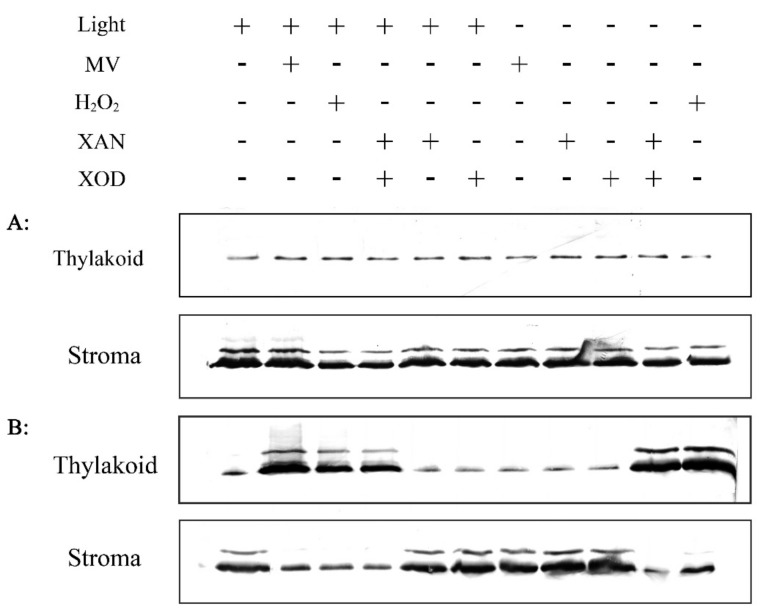
H_2_O_2_ induced association of RCA with the thylakoid membrane. Chloroplasts were isolated from leaves of *O. meyeriana* and then ruptured by osmotic shock. The suspension of ruptured chloroplasts was treated with H_2_O_2_, methyl viologen (MV) or the xanthine (XAN)-xanthine oxidase (XOD) system, before the chloroplasts were sub-fractionated into soluble stroma and thylakoid membrane fractions followed by separation of RCA proteins while using SDS-PAGE. (**A**): Dalixiang, (**B**): *O. meyeriana*.

**Figure 8 plants-08-00351-f008:**
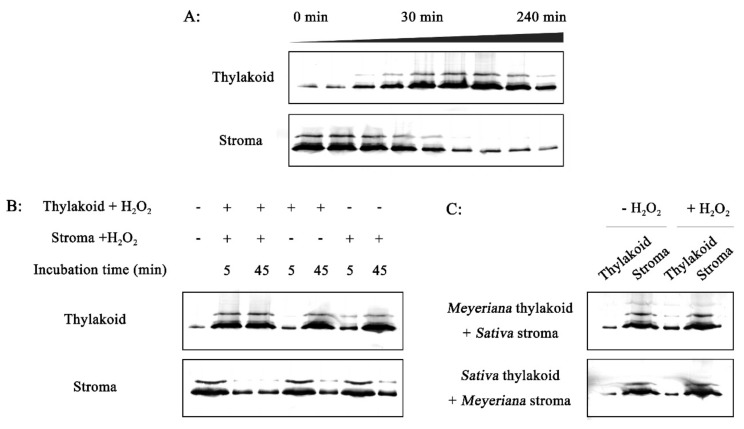
In vitro experiments on RCA association with the thylakoid membrane. (**A**) Effect of different treatment times of exposure to H_2_O_2_ on the association of RCA with the thylakoid membrane of *O. meyeriana*. The suspension of ruptured chloroplasts was treated with H_2_O_2_ during the 240 min. chase phase, and the chloroplasts were sub-fractionated into soluble stroma and thylakoid membrane fractions followed by separation of RCA proteins using SDS-PAGE. (**B**) Effect of pre-treatment with H_2_O_2_ on the stroma or thylakoid membrane fractions of *O. meyeriana*. Intact chloroplasts isolated from leaves were ruptured into soluble stroma fraction and thylakoid membrane fraction, and each of the fractions was treated with H_2_O_2_ for 30 min. before mixing and incubation. At 5 min. or 45 min. after mixing, samples were re-fractionated and the distribution of RCA protein was tested by SDS-PAGE. (**C**) Effect of pre-treatment with H_2_O_2_ as in B but using mixtures of the stroma fraction from susceptible *O. sativa* with the thylakoid membrane fraction of *O. meyeriana* or vice versa.

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
