# Peer review of "H2O2 Induces Association of RCA with the Thylakoid Membrane to Enhance Resistance of Oryza meyeriana to Xanthomonas oryzae pv. oryzae"

_plants, 2019, doi:10.3390/plants8090351_

Round 1
Reviewer 1 Report
This manuscript described characterization of high level of resistance to rice bacterial blight disease by Xanthomonas oryzae pv. oryzae (Xoo) in a wild rice species Oryza meyeriana. The authors reported that Xoo infection induced relocation of Ribrose 1,5-bisphosphate carboxylase/oxygenase (Rubisco) activase (RCA) from chloroplast stroma to thylakoid membrane in O. meyeriana. The authors also investigated radical oxygen burst in infected leaf of O. meyeriana and relationship between the radical oxygen burst and light treatments. However, results in this study are not enough to reach their conclusion, and the authors should describe additional experimental results and discussions in this manuscript described below.
1) The authors described RCA relocation depending on light condition. These results suggest possibility of relationship between the RCA relocation and photoinhibition. Therefore, the author should carry out experiments for chlorophyll fluorescence parameters and Fv/Fm.
2) L.142-146. Did you check mRNA expression of RCA before and after Xoo infection? And, to understand how increasing H2O2 induce Xoo resistance in O meyeriana, the authors should perform Omics analyses such as RNA-seq for investigating gene expressions of other disease resistance-related genes including PR proteins.
3) Fig.4. The CeCl3 assay might be better to investigate subcellular localization of H2O2.
4) L.215-217. This result (moving RVA is light dependent) is important to indicate your significant research. Therefore, I recommend adding explanations about these results at the Abstract section.
5) L.386. The authors should indicate accession No. and origin of O. meyeriana. Is O. meyeriana in this study same with previous studies of reference 3, 4, 12 and 14?
6) L.392. The authors should describe race No. or additional information of Xoo strain used in this study.
7) L.350-351. previous study Han et al. (2015) detected three QTLs for Xoo resistance from O. meyeriana. I would like to know relationship between these QTLs and relocation of RCA to thylakoid membrane. Additional genetic analysis by using O. meyeriana is necessary to identify responsible genes for Xoo resistance, and to understand the relationship between these QTLs and relocation of RCA in O. meyeriana. These descriptions should be included at Discussion section.
8) Fig.1. O. meyeriana differs significantly from Dalixiang in morphology. Large morphological differences such as leaf thickness, length, width and tiller number are not suitable for evaluation and comparison of disease resistance by infection test in O. meyeriana and O. sativa Dalixiang.
Reviewer 2 Report
The article entitled “H2O2 induces association of RCA with the thylakoid membrane to enhance resistance or Oryza meyeriana to Xanthomonas oryzae pv. oryzae” by Mei et al., suggested that the association of RCA with thylakoid membrane is helpful in preventing thylakoid membrane against oxidative damage caused by H2O2 and that H2O2 level increased significantly by Xoo invasion in O. meyeriana leaves.
The paper can be accepted for publication in the Plants but only after following modifications. It requires major change especially in terms of controls for the experiments and confirmation of outcomes.
In Figure 1 – the authors need to quantify the bands in panel B along with they should also show compartment/fraction marker for thylakoid and stroma to confirm that they are not observing any spill over during isolation. Similarly, in Figure 1, panel A they should show the baseline level of H2O2 with H2O control. In Figure 3 – the authors should mention hours in each lane for the corresponding samples. Although I don’t like to suggest new experiments but, in this article, the authors can perform some experiment to show that relationship between increasing H2O2 content is related to relocation of RCA within the chloroplasts and not due to de novo synthesis of RCA. In Figure 4 – panel B, the control with H2O is not very clean as compared to panel A. The authors should mention this in text. There are few typos such as in materials and methods section, line 397 – its not 109 CFU/mL but 109CFU/mL. Authors should check for these typos in the article.Author Response
Please see the attachment.

Round 2
Reviewer 1 Report
Thank you for your revisions of this manuscriptijms-15-11847.
Additional descriptions are reasonable to understand the significance of this study. And, I agree with your comments and reasons that you cannot carry out additional experiments.
Reviewer 2 Report
The authors were able to reply all the queries and the revised manuscript can be accepted in the present form.